# How Does Batch Normalization Help Optimization?

**Shibani Santurkar**[*]
MIT
shibani@mit.edu

**Dimitris Tsipras**[*]
MIT
tsipras@mit.edu

**Andrew Ilyas**[*]
MIT
ailyas@mit.edu

**Aleksander Mądry**
MIT
madry@mit.edu

## Abstract

Batch Normalization (BatchNorm) is a widely adopted technique that enables faster and more stable training of deep neural networks (DNNs). Despite its pervasiveness, the exact reasons for BatchNorm's effectiveness are still poorly understood. The popular belief is that this effectiveness stems from controlling the change of the layers' input distributions during training to reduce the so-called "internal covariate shift". In this work, we demonstrate that such distributional stability of layer inputs has little to do with the success of BatchNorm. Instead, we uncover a more fundamental impact of BatchNorm on the training process: it makes the optimization landscape significantly smoother. This smoothness induces a more predictive and stable behavior of the gradients, allowing for faster training.

## 1 Introduction

Over the last decade, deep learning has made impressive progress on a variety of notoriously difficult tasks in computer vision [16, 7], speech recognition [5], machine translation [29], and game-playing [18, 25]. This progress hinged on a number of major advances in terms of hardware, datasets [15, 23], and algorithmic and architectural techniques [27, 12, 20, 28]. One of the most prominent examples of such advances was batch normalization (BatchNorm) [10].

At a high level, BatchNorm is a technique that aims to improve the training of neural networks by stabilizing the distributions of layer inputs. This is achieved by introducing additional network layers that control the first two moments (mean and variance) of these distributions.

The practical success of BatchNorm is indisputable. By now, it is used by default in most deep learning models, both in research (more than 6,000 citations) and real-world settings. Somewhat shockingly, however, despite its prominence, we still have a poor understanding of what the effectiveness of BatchNorm is stemming from. In fact, there are now a number of works that provide alternatives to BatchNorm [1, 3, 13, 31], but none of them seem to bring us any closer to understanding this issue. (A similar point was also raised recently in [22].)

Currently, the most widely accepted explanation of BatchNorm's success, as well as its original motivation, relates to so-called *internal covariate shift* (ICS). Informally, ICS refers to the change in the distribution of layer inputs caused by updates to the preceding layers. It is conjectured that such continual change negatively impacts training. The goal of BatchNorm was to reduce ICS and thus remedy this effect.

Even though this explanation is widely accepted, we seem to have little concrete evidence supporting it. In particular, we still do not understand the link between ICS and training performance. The chief goal of this paper is to address all these shortcomings. Our exploration lead to somewhat startling discoveries.

---

[*]Equal contribution.

**Our Contributions.** Our point of start is demonstrating that there does not seem to be any link between the performance gain of BatchNorm and the reduction of internal covariate shift. Or that this link is tenuous, at best. In fact, we find that in a certain sense *BatchNorm might not even be reducing internal covariate shift*.

We then turn our attention to identifying the roots of BatchNorm's success. Specifically, we demonstrate that BatchNorm impacts network training in a fundamental way: it makes the landscape of the corresponding optimization problem *significantly more smooth*. This ensures, in particular, that the gradients are more predictive and thus allows for use of larger range of learning rates and faster network convergence. We provide an empirical demonstration of these findings as well as their theoretical justification. We prove that, under natural conditions, the Lipschitzness of both the loss and the gradients (also known as $\beta$-smoothness [21]) are improved in models with BatchNorm.

Finally, we find that this smoothening effect is not uniquely tied to BatchNorm. A number of other natural normalization techniques have a similar (and, sometime, even stronger) effect. In particular, they all offer similar improvements in the training performance.

We believe that understanding the roots of such a fundamental techniques as BatchNorm will let us have a significantly better grasp of the underlying complexities of neural network training and, in turn, will inform further algorithmic progress in this context.

Our paper is organized as follows. In Section 2, we explore the connections between BatchNorm, optimization, and internal covariate shift. Then, in Section 3, we demonstrate and analyze the exact roots of BatchNorm's success in deep neural network training. We present our theoretical analysis in Section 4. We discuss further related work in Section 5 and conclude in Section 6.

## 2 Batch normalization and internal covariate shift

Batch normalization (BatchNorm) [10] has been arguably one of the most successful architectural innovations in deep learning. But even though its effectiveness is indisputable, we do not have a firm understanding of why this is the case.

Broadly speaking, BatchNorm is a mechanism that aims to stabilize the distribution (over a mini-batch) of inputs to a given network layer during training. This is achieved by augmenting the network with additional layers that set the first two moments (mean and variance) of the distribution of each activation to be zero and one respectively. Then, the batch normalized inputs are also typically scaled and shifted based on trainable parameters to preserve model expressivity. This normalization is applied before the non-linearity of the previous layer.

One of the key motivations for the development of BatchNorm was the reduction of so-called *internal covariate shift* (ICS). This reduction has been widely viewed as the root of BatchNorm's success. Ioffe and Szegedy [10] describe ICS as the phenomenon wherein the distribution of inputs to a layer in the network changes due to an update of parameters of the previous layers. This change leads to a constant shift of the underlying training problem and is thus believed to have detrimental effect on the training process.

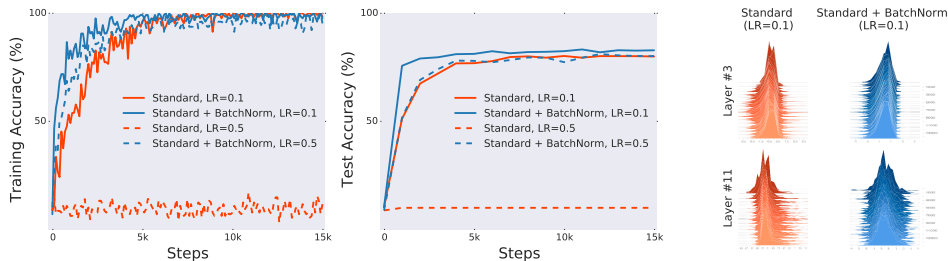

Figure 1: Comparison of (a) training (optimization) and (b) test (generalization) performance of a standard VGG network trained on CIFAR-10 with and without BatchNorm (details in Appendix A). There is a consistent gain in training speed in models with BatchNorm layers. (c) Even though the gap between the performance of the BatchNorm and non-BatchNorm networks is clear, the difference in the evolution of layer input distributions seems to be much less pronounced. (Here, we sampled activations of a given layer and visualized their distribution over training steps.)

Despite its fundamental role and widespread use in deep learning, the underpinnings of BatchNorm's success remain poorly understood [22]. In this work we aim to address this gap. To this end, we start by investigating the connection between ICS and BatchNorm. Specifically, we consider first training a standard VGG [26] architecture on CIFAR-10 [15] with and without BatchNorm. As expected, Figures 1(a) and (b) show a drastic improvement, both in terms of optimization and generalization performance, for networks trained with BatchNorm layers. Figure 1(c) presents, however, a surprising finding. In this figure, we visualize to what extent BatchNorm is stabilizing distributions of layer inputs by plotting the distribution (over a batch) of a random input over training. Surprisingly, the difference in distributional stability (change in the mean and variance) in networks with and without BatchNorm layers seems to be marginal. This observation raises the following questions:

(1) *Is the effectiveness of BatchNorm indeed related to internal covariate shift?*
(2) *Is BatchNorm's stabilization of layer input distributions even effective in reducing ICS?*

We now explore these questions in more depth.

## 2.1    Does BatchNorm's performance stem from controlling internal covariate shift?

The central claim in [10] is that controlling the mean and variance of distributions of layer inputs is directly connected to improved training performance. Can we, however, substantiate this claim?

We propose the following experiment. We train networks with *random* noise injected *after* BatchNorm layers. Specifically, we perturb each activation for each sample in the batch using i.i.d. noise sampled from a *non-zero* mean and *non-unit* variance distribution. We emphasize that this noise distribution *changes* at each time step (see Appendix A for implementation details).

Note that such noise injection produces a severe covariate shift that skews activations at every time step. Consequently, every unit in the layer experiences a *different* distribution of inputs at *each* time step. We then measure the effect of this deliberately introduced distributional instability on BatchNorm's performance. Figure 2 visualizes the training behavior of standard, BatchNorm and our "noisy" BatchNorm networks. Distributions of activations over time from layers at the same depth in each one of the three networks are shown alongside.

Observe that the performance difference between models with BatchNorm layers, and "noisy" Batch-Norm layers is almost non-existent. Also, both these networks perform much better than standard networks. Moreover, the "noisy" BatchNorm network has qualitatively *less stable* distributions than even the standard, non-BatchNorm network, yet it *still performs better* in terms of training. To put

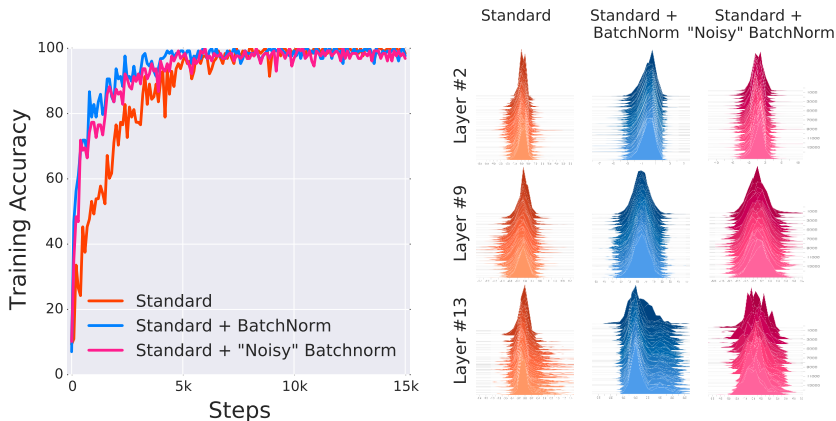

Figure 2: Connections between distributional stability and BatchNorm performance: We compare VGG networks trained without BatchNorm (Standard), with BatchNorm (Standard + BatchNorm) and with explicit "covariate shift" added to BatchNorm layers (Standard + "Noisy" BatchNorm). In the later case, we induce distributional instability by adding *time-varying*, *non-zero* mean and *non-unit* variance noise independently to each batch normalized activation. The "noisy" BatchNorm model nearly matches the performance of standard BatchNorm model, despite complete distributional instability. We sampled activations of a given layer and visualized their distributions (also cf. Figure 7).

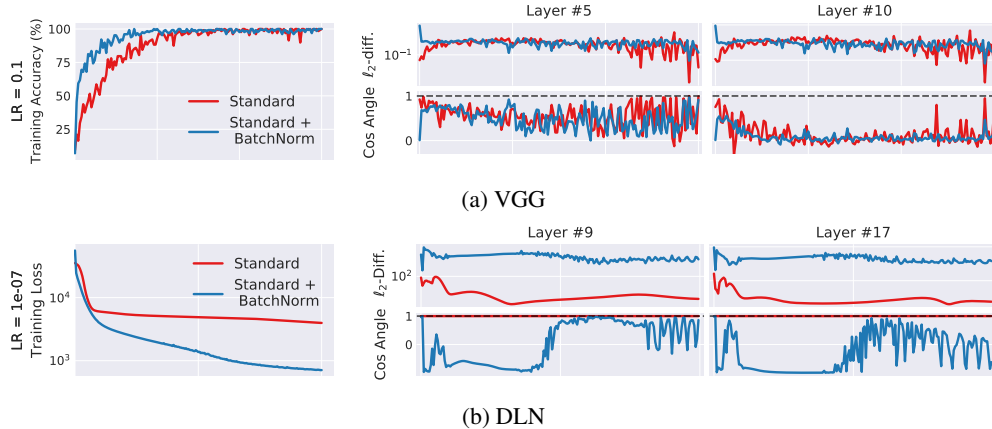

(a) VGG

(b) DLN

Figure 3: Measurement of ICS (as defined in Definition 2.1) in networks with and without BatchNorm layers. For a layer we measure the cosine angle (ideally 1) and $\ell_2$-difference of the gradients (ideally 0) before and after updates to the preceding layers (see Definition 2.1). Models with BatchNorm have similar, or even worse, internal covariate shift, despite performing better in terms of accuracy and loss. (Stabilization of BatchNorm faster during training is an artifact of parameter convergence.)

the magnitude of the noise into perspective, we plot the mean and variance of random activations for select layers in Figure 7. Moreover, adding the same amount of noise to the activations of the standard (non-BatchNorm) network prevents it from training entirely.

Clearly, these findings are hard to reconcile with the claim that the performance gain due to Batch-Norm stems from increased stability of layer input distributions.

## 2.2 Is BatchNorm reducing internal covariate shift?

Our findings in Section 2.1 make it apparent that ICS is not directly connected to the training performance, at least if we tie ICS to stability of the mean and variance of input distributions. One might wonder, however: Is there a broader notion of internal covariate shift that *has* such a direct link to training performance? And if so, does BatchNorm indeed reduce this notion?

Recall that each layer can be seen as solving an empirical risk minimization problem where given a set of inputs, it is optimizing some loss function (that possibly involves later layers). An update to the parameters of any previous layer will change these inputs, thus changing this empirical risk minimization problem itself. This phenomenon is at the core of the intuition that Ioffe and Szegedy [10] provide regarding internal covariate shift. Specifically, they try to capture this phenomenon from the perspective of the resulting *distributional* changes in layer inputs. However, as demonstrated in Section 2.1, this perspective does not seem to properly encapsulate the roots of BatchNorm's success.

To answer this question, we consider a broader notion of internal covariate shift that is more tied to the underlying optimization task. (After all the success of BatchNorm is largely of an optimization nature.) Since the training procedure is a first-order method, the gradient of the loss is the most natural object to study. To quantify the extent to which the parameters in a layer would have to "adjust" in reaction to a parameter update in the previous layers, we measure the difference between the gradients of each layer before and after updates to all the previous layers. This leads to the following definition.

**Definition 2.1.** *Let $\mathcal{L}$ be the loss, $W_1^{(t)}, \ldots, W_k^{(t)}$ be the parameters of each of the $k$ layers and $(x^{(t)}, y^{(t)})$ be the batch of input-label pairs used to train the network at time $t$. We define* internal covariate shift *(ICS) of activation $i$ at time $t$ to be the difference $||G_{t,i} - G'_{t,i}||_2$, where*

$$G_{t,i} = \nabla_{W_i^{(t)}} \mathcal{L}(W_1^{(t)}, \ldots, W_k^{(t)}; x^{(t)}, y^{(t)})$$

$$G'_{t,i} = \nabla_{W_i^{(t)}} \mathcal{L}(W_1^{(t+1)}, \ldots, W_{i-1}^{(t+1)}, W_i^{(t)}, W_{i+1}^{(t)}, \ldots, W_k^{(t)}; x^{(t)}, y^{(t)}).$$

Here, $G_{t,i}$ corresponds to the gradient of the layer parameters that would be applied during a simultaneous update of all layers (as is typical). On the other hand, $G'_{t,i}$ is the same gradient *after* all

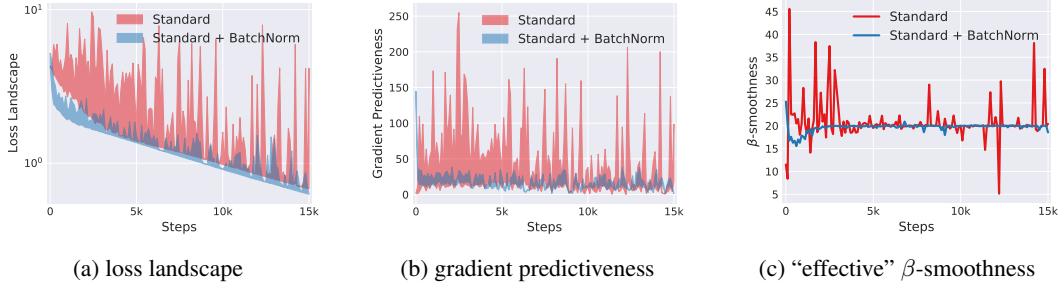

| (a) loss landscape | (b) gradient predictiveness | (c) "effective" $\beta$-smoothness |

Figure 4: Analysis of the optimization landscape of VGG networks. At a particular training step, we measure the variation (shaded region) in loss (a) and $\ell_2$ changes in the gradient (b) as we move in the gradient direction. The "effective" $\beta$-smoothness (c) refers to the maximum difference (in $\ell_2$-norm) in gradient over distance moved in that direction. There is a clear improvement in all of these measures in networks with BatchNorm, indicating a more well-behaved loss landscape. (Here, we cap the maximum distance to be $\eta = 0.4\times$ the gradient since for larger steps the standard network just performs worse (see Figure 1). BatchNorm however continues to provide smoothing for even larger distances.) Note that these results are supported by our theoretical findings (Section 4).

the previous layers have been updated with their new values. The difference between $G$ and $G'$ thus reflects the change in the optimization landscape of $W_i$ caused by the changes to its input. It thus captures precisely the effect of cross-layer dependencies that could be problematic for training.

Equipped with this definition, we measure the extent of ICS with and without BatchNorm layers. To isolate the effect of non-linearities as well as gradient stochasticity, we also perform this analysis on (25-layer) deep linear networks (DLN) trained with full-batch gradient descent (see Appendix A for details). The conventional understanding of BatchNorm suggests that the addition of BatchNorm layers in the network should increase the correlation between $G$ and $G'$, thereby reducing ICS.

Surprisingly, we observe that networks with BatchNorm often exhibit an *increase* in ICS (cf. Figure 3). This is particularly striking in the case of DLN. In fact, in this case, the standard network experiences almost no ICS for the entirety of training, whereas for BatchNorm it appears that $G$ and $G'$ are almost uncorrelated. We emphasize that this is the case *even though BatchNorm networks continue to perform drastically better* in terms of attained accuracy and loss. (The stabilization of the BatchNorm VGG network later in training is an artifact of faster convergence.) This evidence suggests that, from optimization point of view BatchNorm might not even reduce the internal covariate shift.

# 3 Why does BatchNorm work?

Our investigation so far demonstrated that the generally asserted link between the internal covariate shift (ICS) and the optimization performance is tenuous, at best. But BatchNorm *does* significantly improve the training process. Can we explain why this is the case?

Aside from reducing ICS, Ioffe and Szegedy [10] identify a number of additional properties of BatchNorm. These include prevention of exploding or vanishing gradients, robustness to different settings of hyperparameters such as learning rate and initialization scheme, and keeping most of the activations away from saturation regions of non-linearities. All these properties are clearly beneficial to the training process. But they are fairly simple consequences of the mechanics of BatchNorm and do little to uncover the underlying factors responsible for BatchNorm's success. *Is there a more fundamental phenomenon at play here?*

## 3.1 The smoothing effect of BatchNorm

Indeed, we identify the key impact that BatchNorm has on the training process: it reparametrizes the underlying optimization problem to *make its landscape significantly more smooth*. The first manifestation of this impact is improvement in the Lipschitzness[2] of the loss function. That is, the loss changes at a smaller rate and the magnitudes of the gradients are smaller too. There is, however,

an even stronger effect at play. Namely, BatchNorm's reparametrization makes *gradients* of the loss more Lipschitz too. In other words, the loss exhibits a significantly better "effective" $\beta$-smoothness[3].

These smoothening effects impact the performance of the training algorithm in a major way. To understand why, recall that in a vanilla (non-BatchNorm), deep neural network, the loss function is not only non-convex but also tends to have a large number of "kinks", flat regions, and sharp minima [17]. This makes gradient descent–based training algorithms unstable, e.g., due to exploding or vanishing gradients, and thus highly sensitive to the choice of the learning rate and initialization.

Now, the key implication of BatchNorm's reparametrization is that it makes the gradients more reliable and predictive. After all, improved Lipschitzness of the gradients gives us confidence that when we take a larger step in a direction of a computed gradient, this gradient direction remains a fairly accurate estimate of the actual gradient direction after taking that step. It thus enables any (gradient–based) training algorithm to take larger steps without the danger of running into a sudden change of the loss landscape such as flat region (corresponding to vanishing gradient) or sharp local minimum (causing exploding gradients). This, in turn, enables us to use a broader range of (and thus larger) learning rates (see Figure 10 in Appendix B) and, in general, makes the training significantly faster and less sensitive to hyperparameter choices. (This also illustrates how the properties of BatchNorm that we discussed earlier can be viewed as a manifestation of this smoothening effect.)

## 3.2 Exploration of the optimization landscape

To demonstrate the impact of BatchNorm on the stability of the loss itself, i.e., its Lipschitzness, for each given step in the training process, we compute the gradient of the loss at that step and measure how the loss changes as we move in that direction – see Figure 4(a). We see that, in contrast to the case when BatchNorm is in use, the loss of a vanilla, i.e., non-BatchNorm, network has a very wide range of values along the direction of the gradient, especially in the initial phases of training. (In the later stages, the network is already close to convergence.)

Similarly, to illustrate the increase in the stability and predictiveness of the gradients, we make analogous measurements for the $\ell_2$ distance between the loss gradient at a given point of the training and the gradients corresponding to different points along the original gradient direction. Figure 4(b) shows a significant difference (close to two orders of magnitude) in such gradient predictiveness between the vanilla and BatchNorm networks, especially early in training.

To further demonstrate the effect of BatchNorm on the stability/Lipschitzness of the gradients of the loss, we plot in Figure 4(c) the "effective" $\beta$-smoothness of the vanilla and BatchNorm networks throughout the training. ("Effective" refers here to measuring the change of gradients as we move in the direction of the gradients.). Again, we observe consistent differences between these networks. We complement the above examination by considering *linear* deep networks: as shown in Figures 9 and 12 in Appendix B, the BatchNorm smoothening effect is present there as well.

Finally, we emphasize that even though our explorations were focused on the behavior of the loss along the gradient directions (as they are the crucial ones from the point of view of the training process), the loss behaves in a similar way when we examine other (random) directions too.

## 3.3 Is BatchNorm the best (only?) way to smoothen the landscape?

Given this newly acquired understanding of BatchNorm and the roots of its effectiveness, it is natural to wonder: *Is this smoothening effect a unique feature of BatchNorm?* Or could a similar effect be achieved using some other normalization schemes?

To answer this question, we study a few natural data statistics-based normalization strategies. Specifically, we study schemes that fix the first order moment of the activations, as BatchNorm does, and then normalizes them by the average of their $\ell_p$-norm (*before* shifting the mean), for $p = 1, 2, \infty$. Note that for these normalization schemes, the distributions of layer inputs are no longer Gaussian-like (see Figure 14). Hence, normalization with such $\ell_p$-norm does not guarantee anymore any control over the distribution moments nor distributional stability.

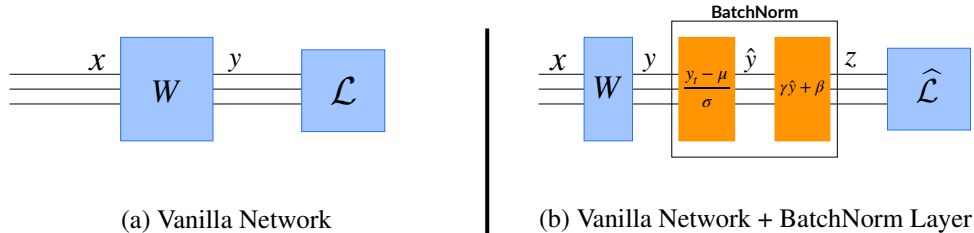

(a) Vanilla Network | (b) Vanilla Network + BatchNorm Layer

Figure 5: The two network architectures we compare in our theoretical analysis: (a) the vanilla DNN (no BatchNorm layer); (b) the same network as in (a) but with a BatchNorm layer inserted after the fully-connected layer $W$. (All the layer parameters have exactly the same value in both networks.)

The results are presented in Figures 13, 11 and 12 in Appendix B. We observe that all the normalization strategies offer comparable performance to BatchNorm. In fact, for deep linear networks, $\ell_1-$ normalization performs even better than BatchNorm. Note that, qualitatively, the $\ell_p-$normalization techniques lead to *larger distributional shift* (as considered in [10]) than the vanilla, i.e., unnormalized, networks, yet they still *yield improved optimization performance*. Also, all these techniques result in an improved smoothness of the landscape that is similar to the effect of BatchNorm. (See Figures 11 and 12 of Appendix B.) This suggests that the positive impact of BatchNorm on training might be somewhat serendipitous. Therefore, it might be valuable to perform a principled exploration of the design space of normalization schemes as it can lead to better performance.

## 4 Theoretical Analysis

Our experiments so far suggest that BatchNorm has a fundamental effect on the optimization landscape. We now explore this phenomenon from a theoretical perspective. To this end, we consider an arbitrary linear layer in a DNN (we do not necessitate that the entire network be fully linear).

### 4.1 Setup

We analyze the impact of adding a single BatchNorm layer after an arbitrary fully-connected layer $W$ at a given step during the training. Specifically, we compare the optimization landscape of the original training problem to the one that results from inserting the BatchNorm layer *after* the fully-connected layer – normalizing the output of this layer (see Figure 5). Our analysis therefore captures effects that stem from the reparametrization of the landscape and not merely from normalizing the inputs $x$.

We denote the layer weights (identical for both the standard and batch-normalized networks) as $W_{ij}$. Both networks have the same arbitrary loss function $\mathcal{L}$ that could potentially include a number of additional non-linear layers after the current one. We refer to the loss of the normalized network as $\widehat{\mathcal{L}}$ for clarity. In both networks, we have input $x$, and let $y = Wx$. For networks with BatchNorm, we have an additional set of activations $\hat{y}$, which are the "whitened" version of $y$, i.e. standardized to mean 0 and variance 1. These are then multiplied by $\gamma$ and added to $\beta$ to form $z$. We assume $\beta$ and $\gamma$ to be constants for our analysis. In terms of notation, we let $\sigma_j$ denote the standard deviation (computed over the mini-batch) of a batch of outputs $y_j \in \mathbb{R}^m$.

### 4.2 Theoretical Results

We begin by considering the optimization landscape with respect to the activations $\boldsymbol{y_j}$. We show that batch normalization causes this landscape to be more well-behaved, inducing favourable properties in Lipschitz-continuity, and predictability of the gradients. We then show that these improvements in the activation-space landscape translate to favorable worst-case bounds in the weight-space landscape.

We first turn our attention to the gradient magnitude $\left|\left|\nabla_{\boldsymbol{y_j}} \mathcal{L}\right|\right|$, which captures the Lipschitzness of the loss. The Lipschitz constant of the loss plays a crucial role in optimization, since it controls the amount by which the loss can change when taking a step (see [21] for details). Without any assumptions on the specific weights or the loss being used, we show that the batch-normalized

landscape exhibits a better Lipschitz constant. Moreover, the Lipschitz constant is significantly reduced whenever the activations $\hat{\boldsymbol{y}}_j$ correlate with the gradient $\nabla_{\hat{\boldsymbol{y}}_j}\widehat{\mathcal{L}}$ or the mean of the gradient deviates from $0$. Note that this reduction is additive, and has effect even when the scaling of BN is identical to the original layer scaling (i.e. even when $\sigma_j = \gamma$).

**Theorem 4.1** (The effect of BatchNorm on the Lipschitzness of the loss). *For a BatchNorm network with loss $\widehat{\mathcal{L}}$ and an identical non-BN network with (identical) loss $\mathcal{L}$,*

$$\left\| \nabla_{\boldsymbol{y}_j} \widehat{\mathcal{L}} \right\|^2 \le \frac{\gamma^2}{\sigma_j^2} \left( \left\| \nabla_{\boldsymbol{y}_j} \mathcal{L} \right\|^2 - \frac{1}{m} \left\langle \mathbf{1}, \nabla_{\boldsymbol{y}_j} \mathcal{L} \right\rangle^2 - \frac{1}{\sqrt{m}} \left\langle \nabla_{\boldsymbol{y}_j} \mathcal{L}, \hat{\boldsymbol{y}}_j \right\rangle^2 \right).$$

First, note that $\langle \mathbf{1}, \partial L/\partial y \rangle^2$ grows quadratically in the dimension, so the middle term above is significant. Furthermore, the final inner product term is expected to be bounded away from zero, as the gradient with respect to a variable is rarely uncorrelated to the variable itself. In addition to the additive reduction, $\sigma_j$ tends to be large in practice (cf. Appendix Figure 8), and thus the scaling by $\frac{\gamma}{\sigma}$ may contribute to the relative "flatness" we see in the effective Lipschitz constant.

We now turn our attention to the second-order properties of the landscape. We show that when a BatchNorm layer is added, the quadratic form of the loss Hessian with respect to the activations in the gradient direction, is both rescaled by the input variance (inducing resilience to mini-batch variance), and decreased by an additive factor (increasing smoothness). This term captures the second order term of the Taylor expansion of the gradient around the current point. Therefore, reducing this term implies that the first order term (the gradient) is more predictive.

**Theorem 4.2** (The effect of BN to smoothness). *Let $\hat{\boldsymbol{g}}_j = \nabla_{\boldsymbol{y}_j} \mathcal{L}$ and $\boldsymbol{H}_{jj} = \frac{\partial \mathcal{L}}{\partial \boldsymbol{y}_j \partial \boldsymbol{y}_j}$ be the gradient and Hessian of the loss with respect to the layer outputs respectively. Then*

$$\left( \nabla_{\boldsymbol{y}_j} \widehat{\mathcal{L}} \right)^\top \frac{\partial \widehat{\mathcal{L}}}{\partial \boldsymbol{y}_j \partial \boldsymbol{y}_j} \left( \nabla_{\boldsymbol{y}_j} \widehat{\mathcal{L}} \right) \le \frac{\gamma^2}{\sigma^2} \left( \frac{\partial \widehat{\mathcal{L}}}{\partial \boldsymbol{y}_j} \right)^\top \boldsymbol{H}_{jj} \left( \frac{\partial \widehat{\mathcal{L}}}{\partial \boldsymbol{y}_j} \right) - \frac{\gamma}{m\sigma^2} \langle \hat{\boldsymbol{g}}_j, \hat{\boldsymbol{y}}_j \rangle \left\| \frac{\partial \widehat{\mathcal{L}}}{\partial \boldsymbol{y}_j} \right\|^2$$

*If we also have that the $\boldsymbol{H}_{jj}$ preserves the relative norms of $\hat{\boldsymbol{g}}_j$ and $\nabla_{\boldsymbol{y}_j} \widehat{\mathcal{L}}$,*

$$\left( \nabla_{\boldsymbol{y}_j} \widehat{\mathcal{L}} \right)^\top \frac{\partial \widehat{\mathcal{L}}}{\partial \boldsymbol{y}_j \partial \boldsymbol{y}_j} \left( \nabla_{\boldsymbol{y}_j} \widehat{\mathcal{L}} \right) \le \frac{\gamma^2}{\sigma^2} \left( \hat{\boldsymbol{g}}_j^\top \boldsymbol{H}_{jj} \hat{\boldsymbol{g}}_j - \frac{1}{m\gamma} \langle \hat{\boldsymbol{g}}_j, \hat{\boldsymbol{y}}_j \rangle \left\| \frac{\partial \widehat{\mathcal{L}}}{\partial \boldsymbol{y}_j} \right\|^2 \right)$$

Note that if the quadratic forms involving the Hessian and the inner product $\langle \hat{\boldsymbol{y}}_j, \hat{\boldsymbol{g}}_j \rangle$ are non-negative (both fairly mild assumptions), the theorem implies more predictive gradients. The Hessian is positive semi-definite (PSD) if the loss is locally convex which is true for the case of deep networks with piecewise-linear activation functions and a convex loss at the final layer (e.g. standard softmax cross-entropy loss or other common losses). The condition $\langle \hat{\boldsymbol{y}}_j, \hat{\boldsymbol{g}}_j \rangle > 0$ holds as long as the negative gradient $\hat{\boldsymbol{g}}_j$ is pointing towards the minimum of the loss (w.r.t. normalized activations). Overall, as long as these two conditions hold, the steps taken by the BatchNorm network are more predictive than those of the standard network (similarly to what we observed experimentally).

Note that our results stem from the reparametrization of the problem and not a simple scaling.

**Observation 4.3** (BatchNorm does more than rescaling). *For any input data $X$ and network configuration $W$, there exists a BN configuration $(W, \gamma, \beta)$ that results in the same activations $y_j$, and where $\gamma = \sigma_j$. Consequently, all of the minima of the normal landscape are preserved in the BN landscape.*

Our theoretical analysis so far studied the optimization landscape of the loss w.r.t. the normalized activations. We will now translate these bounds to a favorable worst-case bound on the landscape with respect to layer weights. Note that a (near exact) analogue of this theorem for minimax gradient predictiveness appears in Theorem C.1 of Appendix C.

**Theorem 4.4** (Minimax bound on weight-space Lipschitzness). *For a BatchNorm network with loss $\widehat{\mathcal{L}}$ and an identical non-BN network (with identical loss $\mathcal{L}$), if*

$$g_j = \max_{\|X\| \le \lambda} \|\nabla_W \mathcal{L}\|^2, \qquad \hat{g}_j = \max_{\|X\| \le \lambda} \left\| \nabla_W \widehat{\mathcal{L}} \right\|^2 \implies \hat{g}_j \le \frac{\gamma^2}{\sigma_j^2} \left( g_j^2 - m\mu_{g_j}^2 - \lambda^2 \left\langle \nabla_{\boldsymbol{y}_j} \mathcal{L}, \hat{\boldsymbol{y}}_j \right\rangle^2 \right).$$

Finally, in addition to a desirable landscape, we find that BN also offers an advantage in initialization:

**Lemma 4.5** (BatchNorm leads to a favourable initialization). *Let $W^*$ and $\widehat{W}^*$ be the set of local optima for the weights in the normal and BN networks, respectively. For any initialization $W_0$*

$$\left\| \left| W_0 - \widehat{W}^* \right| \right\|^2 \leq ||W_0 - W^*||^2 - \frac{1}{||W^*||^2} \left( ||W^*||^2 - \langle W^*, W_0 \rangle \right)^2,$$

*if $\langle W_0, W^* \rangle > 0$, where $\widehat{W}^*$ and $W^*$ are closest optima for BN and standard network, respectively.*

## 5  Related work

A number of normalization schemes have been proposed as alternatives to BatchNorm, including normalization over layers [1], subsets of the batch [31], or across image dimensions [30]. Weight Normalization [24] follows a complementary approach normalizing the weights instead of the activations. Finally, ELU [3] and SELU [13] are two proposed examples of non-linearities that have a progressively decaying slope instead of a sharp saturation and can be used as an alternative for BatchNorm. These techniques offer an improvement over standard training that is comparable to that of BatchNorm but do not attempt to explain BatchNorm's success.

Additionally, work on topics related to DNN optimization has uncovered a number of other Batch-Norm benefits. Li et al. [9] observe that networks with BatchNorm tend to have optimization trajectories that rely less on the parameter initialization. Balduzzi et al. [2] observe that models without BatchNorm tend to suffer from small correlation between different gradient coordinates and/or unit activations. They report that this behavior is profound in deeper models and argue how it constitutes an obstacle to DNN optimization. Morcos et al. [19] focus on the generalization properties of DNN. They observe that the use of BatchNorm results in models that rely less on single directions in the activation space, which they find to be connected to the generalization properties of the model.

Recent work [14] identifies simple, concrete settings where a variant of training with BatchNorm provably improves over standard training algorithms. The main idea is that decoupling the length and direction of the weights (as done in BatchNorm and Weight Normalization [24]) can be exploited to a large extent. By designing algorithms that optimize these parameters separately, with (different) adaptive step sizes, one can achieve significantly faster convergence rates for these problems.

## 6  Conclusions

In this work, we have investigated the roots of BatchNorm's effectiveness as a technique for training deep neural networks. We find that the widely believed connection between the performance of BatchNorm and the internal covariate shift is tenuous, at best. In particular, we demonstrate that existence of internal covariate shift, at least when viewed from the – generally adopted – distributional stability perspective, is *not* a good predictor of training performance. Also, we show that, from an optimization viewpoint, BatchNorm might not be even reducing that shift.

Instead, we identify a key effect that BatchNorm has on the training process: it reparametrizes the underlying optimization problem to make it more stable (in the sense of loss Lipschitzness) and smooth (in the sense of "effective" $\beta$-smoothness of the loss). This implies that the gradients used in training are more predictive and well-behaved, which enables faster and more effective optimization. This phenomena also explains and subsumes some of the other previously observed benefits of BatchNorm, such as robustness to hyperparameter setting and avoiding gradient explosion/vanishing. We also show that this smoothing effect is not unique to BatchNorm. In fact, several other natural normalization strategies have similar impact and result in a comparable performance gain.

We believe that these findings not only challenge the conventional wisdom about BatchNorm but also bring us closer to a better understanding of this technique. We also view these results as an opportunity to encourage the community to pursue a more systematic investigation of the algorithmic toolkit of deep learning and the underpinnings of its effectiveness.

Finally, our focus here was on the impact of BatchNorm on training but our findings might also shed some light on the BatchNorm's tendency to improve generalization. Specifically, it could be the case that the smoothening effect of BatchNorm's reparametrization encourages the training process to converge to more flat minima. Such minima are believed to facilitate better generalization [8, 11]. We hope that future work will investigate this intriguing possibility.

## Acknowledgements

We thank Ali Rahimi and Ben Recht for helpful comments on a preliminary version of this paper.

Shibani Santurkar was supported by the National Science Foundation (NSF) under grants IIS-1447786, IIS-1607189, and CCF-1563880, and the Intel Corporation. Dimitris Tsipras was supported in part by the NSF grant CCF-1553428 and the NSF Frontier grant CNS-1413920. Andrew Ilyas was supported in part by NSF awards CCF-1617730 and IIS-1741137, a Simons Investigator Award, a Google Faculty Research Award, and an MIT-IBM Watson AI Lab research grant. Aleksander Mądry was supported in part by an Alfred P. Sloan Research Fellowship, a Google Research Award, and the NSF grants CCF-1553428 and CNS-1815221.

## Footnotes

[2]Recall that $f$ is $L$-Lipschitz if $|f(x_1) - f(x_2)| \le L\|x_1 - x_2\|$, for all $x_1$ and $x_2$.

[3]Recall that $f$ is $\beta$-smooth if its gradient is $\beta$-Lipschitz. It is worth noting that, due to the existence of non-linearities, one should not expect the $\beta$-smoothness to be bounded in an absolute, global sense.

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
