[Supplementary Material]

# A Experimental Setup

In Section A.1, we provide details regarding the architectures used in our analysis. Then in Section A.2 we discuss the specifics of the setup and measurements used in our experiments.

## A.1 Models

We use two standard deep architectures – a VGG-like network, and a deep *linear* network (DLN). The VGG model achieves close to state-of-the-art performance while being fairly simple[4]. Preliminary experiments on other architectures gave similar results. We study DLNs with full-batch training since they allow us to isolate the effect of non-linearities, as well as the stochasticity of the training procedure. Both these architectures show clear a performance benefits with BatchNorm.

Specific details regarding both architectures are provided below:

1. **Convolutional VGG architecture on CIFAR10 (VGG):**

   We fit a VGG-like network, a standard convolutional architecture [26], to a canonical image classification problem (CIFAR10 [15]). We optimize using standard stochastic gradient descent and train for $15,000$ steps (training accuracy plateaus). We use a batch size of $128$ and a fixed learning rate of $0.1$ unless otherwise specified. Moreover, since our focus is on training, we do not use data augmentation. This architecture can fit the training dataset well and achieves close to state-of-the art test performance. Our network achieves a test accuracy of 83% with BatchNorm and 80% without (this becomes 92% and 88% respectively with data augmentation).

2. $25$**-Layer Deep Linear Network on Synthetic Gaussian Data (DLN):**

   DLN are a factorized approach to solving a simple regression problem, i.e., fitting $Ax$ from $x$. Specifically, we consider a deep network with $k$ fully connected layers and an $\ell_2$ loss. Thus, we are minimizing $\|W_1 \ldots W_k x - Ax\|_2^2$ over $W_i$ [5]. We generate inputs $x$ from a Gaussian Distribution and a matrix $A$ with i.i.d. Gaussian entries. We choose $k$ to be 25, and the dimensions of $A$ to be $10 \times 10$. All the matrices $W_i$ are square and have the same dimensions. We train DLN using full-batch gradient descent for $10,000$ steps (training loss plateaus).The size of the dataset is 1000 (same as the batch size) and the learning rate is $10^{-6}$ unless otherwise specified.

For both networks we use standard Glorot initialization [4]. Further the learning rates were selected based on hyperparameter optimization to find a configuration where the training performance for the network was the best.

## A.2 Details

### A.2.1 "Noisy" BatchNorm Layers

Consider $a_{i,j}$, the $j$-th activation of the $i$-th example in the batch. Note that batch norm will ensure that the distribution of $a_{\cdot,j}$ for some $j$ will have fixed mean and variance (possibly learnable).

At every time step, our noise model consists of perturbing each activation for each sample in a batch with noise i.i.d. from a non-zero mean, non-unit variance distribution $D_j^t$. The distribution $D_j^t$ itself is time varying and its parameters are drawn i.i.d from another distribution $D_j$. The specific noise model is described in Algorithm 1. In our experiments, $n_\mu = 0.5$, $n_\sigma = 1.25$ and $r_\mu = r_\sigma = 0.1$. (For convolutional layers, we follow the standard convention of treating the height and width dimensions as part of the batch.)

**Algorithm 1** "Noisy" BatchNorm

---

1: % **For constants** $n_m$, $n_v$, $r_m$, $r_v$.

2:

3: **for** each layer at time $t$ **do**

4:     $a_{i,j}^t \leftarrow$ *Batch-normalized activation for unit j and sample i*

5:

6:     **for** each $j$ **do**                         ▷ Sample the parameters $(m_j^t, v_j^t)$ of $D_j^t$ from $D_j$

7:         $\mu^t \sim U(-n_\mu, n_\mu)$

8:         $\sigma^t \sim U(1, n_\sigma)$

9:

10:     **for** each $i$ **do**                                       ▷ Sample noise from $D_j^t$

11:         **for** each $j$ **do**

12:             $m_{i,j}^t \sim U(\mu - r_\mu, \mu + r_\mu)$

13:             $s_{i,j}^t \sim \mathcal{N}(\sigma, r_\sigma)$

14:             $a_{i,j}^t \leftarrow s_{i,j}^t \cdot a_{i,j} + m_{i,j}^t$

---

While plotting the distribution of activations, we sample random activations from any given layer of the network and plot its distribution over the batch dimension for fully connected layers, and over the batch, height, width dimension for convolutional layers as is standard convention in BatchNorm for convolutional networks.

### A.2.2 Loss Landscape

To measure the smoothness of the loss landscape of a network during the course of training, we essentially take steps of different lengths in the direction of the gradient and measure the loss values obtained at each step. Note that this is not a training procedure, but an evaluation of the local loss landscape at every step of the training process.

For VGG we consider steps of length ranging from $[1/2, 4] \times$ *step size*, whereas for DLN we choose $[1/100, 30] \times$ *step size*. Here *step size* denotes the hyperparameter setting with which the network is being trained. We choose these ranges to roughly reflect the range of parameters that are valid for standard training of these models. The VGG network is much more sensitive to the learning rate choices (probably due to the non-linearities it includes), so we perform line search over a restricted range of parameters. Further, the maximum step size was chosen slightly smaller than the learning rate at which the standard (no BatchNorm) network diverges during training.

## B   Omitted Figures

Additional visualizations for the analysis performed in Section 3.1 are presented below.

(a) VGG

(b) DLN

Figure 6: Measurement of ICS (as defined in Definition 2.1) in networks with and without BatchNorm layers. For a layer we measure the cosine angle (ideally 1) and $\ell_2$-difference of the gradients (ideally 0) before and after updates to the preceding layers (see Definition 2.1). Models with BatchNorm have similar, or even worse, internal covariate shift, despite performing better in terms of accuracy and loss. (Stabilization of BatchNorm faster during training is an artifact of parameter convergence.)

(a)

(b)

Figure 7: Comparison of change in the first two moments (mean and variance) of distributions of example activations for a given layer between two successive steps of the training process. Here we compare VGG networks trained without BatchNorm (Standard), with BatchNorm (Standard + BatchNorm) and with explicit "covariate shift" added to BatchNorm layers (Standard + "Noisy" BatchNorm). "Noisy" BatchNorm layers have significantly higher ICS than standard networks, yet perform better from an optimization perspective (cf. Figure 2).

Figure 8: Distributions of activations from different layers of a 25-Layer deep linear network. Here we sample a random activation from a given layer to visualize its distribution over training.

(a) loss landscape       (b) gradient predictiveness       (c) "effective" $\beta$-smoothness

Figure 9: Analysis of the optimization landscape during training of deep linear networks with and without BatchNorm. At a particular training step, we measure the variation (shaded region) in loss (a) and $\ell_2$ changes in the gradient (b) as we move in the gradient direction. The "effective" $\beta$-smoothness (c) captures the maximum $\beta$ value observed while moving in this direction. There is a clear improvement in each of these measures of smoothness of the optimization landscape in networks with BatchNorm layers. (Here, we cap the maximum distance moved to be $\eta = 30\times$ the gradient since for larger steps the standard network just performs works (see Figure 1). However, BatchNorm continues to provide smoothing for even larger distances.)

(a) VGG

(b) DLN

Figure 10: Comparison of the predictiveness of gradients with and without BatchNorm. Here, at a given step in the optimization, we measure the $\ell_2$ error between the current gradient, and new gradients which are observed while moving in the direction of the current gradient. We then evaluate how this error varies based on distance traversed in the direction of the gradient. We observe that gradients are significantly more predictive in networks with BatchNorm and change slowly in a given local neighborhood. This explains why networks with BatchNorm are largely robust to a broad range of learning rates.

Figure 11: Evaluation of VGG networks trained with different $\ell_p$ normalization strategies discussed in Section 3.3. *(a):* Comparison of the training performance of the models. *(b, c, d):* Evaluation of the smoothness of optimization landscape in the various models. At a particular training step, we measure the variation (shaded region) in loss (*b*) and $\ell_2$ changes in the gradient (*c*) as we move in the gradient direction. We also measure the maximum $\beta$-smoothness while moving in this direction (*d*). We observe that networks with any normalization strategy have improved performance and smoothness of the loss landscape over standard training.

Figure 12: Evaluation of deep linear networks trained with different $\ell_p$ normalization strategies. We observe that networks with any normalization strategy have improved performance and smoothness of the loss landscape over standard training. Details of the plots are the same as Figure 11 above.

(a) VGG

(b) Deep Linear Model

Figure 13: Evaluation of the training performance of $\ell_p$ normalization techniques discussed in Section 3.3. For both networks, all $\ell_p$ normalization strategies perform comparably or even better than BatchNorm. This indicates that the performance gain with BatchNorm is not about distributional stability (controlling mean and variance).

Figure 14: Activation histograms for the VGG network under different normalizations. Here, we randomly sample activations from a given layer and visualize their distributions. Note that the $\ell_p$-normalization techniques leads to larger distributional covariate shift compared to normal networks, yet yield improved optimization performance.

# C  Proofs

We now prove the stated theorems regarding the landscape induced by batch normalization.

We begin with a few facts that can be derived directly from the closed-form of Batch Normalization, which we use freely in proving the following theorems.

## C.1  Useful facts and setup

We consider the same setup pictured in Figure 5 and described in Section 4.1. Note that in proving the theorems we use partial derivative notation instead of gradient notation, and also rely on a few simple but key facts:

**Fact C.1** (Gradient through BatchNorm). *The gradient $\frac{\partial f}{\partial A^{(b)}}$ through BN and another function $f := f(C)$, where $C = \gamma \cdot B + \beta$, and $B = BN_{0,1}(A) := \frac{A - \mu}{\sigma}$ where $A^{(b)}$ are scalar elements of a batch of size $m$ and variance $\sigma^2$ is*

$$\frac{\partial f}{\partial A^{(b)}} = \frac{\gamma}{m\sigma}\left(m\frac{\partial f}{\partial C^{(b)}} - \sum_{k=1}^{m}\frac{\partial f}{\partial C^{(k)}} - B^{(b)}\sum_{k=1}^{m}\frac{\partial f}{\partial C^{(k)}}B^{(k)}\right)$$

**Fact C.2** (Gradients of normalized outputs). *A convenient gradient of BN is given as*

$$\frac{\partial \hat{y}^{(b)}}{\partial y^{(k)}} = \frac{1}{\sigma}\left(\boldsymbol{I}[b = k] - \frac{1}{m} - \frac{1}{m}\hat{y}^{(b)}\hat{y}^{(k)}\right), \tag{1}$$

*and thus*

$$\frac{\partial z_j^{(b)}}{\partial y^{(k)}} = \frac{\gamma}{\sigma}\left(\boldsymbol{I}[b = k] - \frac{1}{m} - \frac{1}{m}\hat{y}^{(b)}\hat{y}^{(k)}\right), \tag{2}$$

## C.2 Lipschitzness proofs

Now, we provide a proof for the Lipschitzness of the loss landscape in terms of the layer activations. In particular, we prove the following theorem from Section 4.

**Theorem 4.1** (The effect of BatchNorm on the Lipschitzness of the loss). *For a BatchNorm network with loss $\widehat{\mathcal{L}}$ and an identical non-BN network with (identical) loss $\mathcal{L}$,*

$$\left\| \nabla_{\boldsymbol{y_j}} \widehat{\mathcal{L}} \right\|^2 \leq \frac{\gamma^2}{\sigma_j^2} \left( ||\nabla_{\boldsymbol{y_j}} \mathcal{L}||^2 - \frac{1}{m} \left\langle \boldsymbol{1}, \nabla_{\boldsymbol{y_j}} \mathcal{L} \right\rangle^2 - \frac{1}{\sqrt{m}} \left\langle \nabla_{\boldsymbol{y_j}} \mathcal{L}, \hat{\boldsymbol{y}}_j \right\rangle^2 \right).$$

*Proof.* Proving this is simply a direct application of Fact C.1. In particular, we have that

$$\frac{\partial \widehat{\mathcal{L}}}{\partial y_j{}^{(b)}} = \left( \frac{\gamma}{m\sigma_j} \right) \left( m \frac{\partial \widehat{\mathcal{L}}}{\partial z_j{}^{(b)}} - \sum_{k=1}^{m} \frac{\partial \widehat{\mathcal{L}}}{\partial z_j{}^{(k)}} - \hat{y}_j^{(b)} \sum_{k=1}^{m} \frac{\partial \widehat{\mathcal{L}}}{\partial z_j{}^{(k)}} \hat{y}_j^{(k)} \right), \tag{3}$$

which we can write in vectorized form as

$$\frac{\partial \widehat{\mathcal{L}}}{\partial \boldsymbol{y_j}} = \left( \frac{\gamma}{m\sigma_j} \right) \left( m \frac{\partial \widehat{\mathcal{L}}}{\partial \boldsymbol{z_j}} - \boldsymbol{1} \left\langle \boldsymbol{1}, \frac{\partial \widehat{\mathcal{L}}}{\partial \boldsymbol{z_j}} \right\rangle - \hat{\boldsymbol{y}}_j \left\langle \frac{\partial \widehat{\mathcal{L}}}{\partial \boldsymbol{z_j}}, \hat{\boldsymbol{y}}_j \right\rangle \right) \tag{4}$$

Now, let $\mu_g = \frac{1}{m} \left\langle \boldsymbol{1}, \partial \widehat{\mathcal{L}}/\partial \boldsymbol{z_j} \right\rangle$ be the mean of the gradient vector, we can rewrite the above as the following (in the subsequent steps taking advantage of the fact that $\hat{\boldsymbol{y}}_j$ is mean-zero and norm-$\sqrt{m}$:

$$\frac{\partial \widehat{\mathcal{L}}}{\partial \boldsymbol{y_j}} = \left( \frac{\gamma}{\sigma_j} \right) \left( \left( \frac{\partial \widehat{\mathcal{L}}}{\partial \boldsymbol{z_j}} - \boldsymbol{1}\mu_g \right) - \frac{1}{m} \hat{\boldsymbol{y}}_j \left\langle \left( \frac{\partial \widehat{\mathcal{L}}}{\partial \boldsymbol{z_j}} - \boldsymbol{1}\mu_g \right), \hat{\boldsymbol{y}}_j \right\rangle \right) \tag{5}$$

$$= \frac{\gamma}{\sigma} \left( \left( \frac{\partial \widehat{\mathcal{L}}}{\partial \boldsymbol{z_j}} - \boldsymbol{1}\mu_g \right) - \frac{\hat{\boldsymbol{y}}_j}{||\hat{\boldsymbol{y}}_j||} \left\langle \left( \frac{\partial \widehat{\mathcal{L}}}{\partial \boldsymbol{z_j}} - \boldsymbol{1}\mu_g \right), \frac{\hat{\boldsymbol{y}}_j}{||\hat{\boldsymbol{y}}_j||} \right\rangle \right) \tag{6}$$

$$\left\| \frac{\partial \widehat{\mathcal{L}}}{\partial \boldsymbol{y_j}} \right\|^2 = \frac{\gamma^2}{\sigma^2} \left\| \left( \frac{\partial \widehat{\mathcal{L}}}{\partial \boldsymbol{z_j}} - \boldsymbol{1}\mu_g \right) - \frac{\hat{\boldsymbol{y}}_j}{||\hat{\boldsymbol{y}}_j||} \left\langle \left( \frac{\partial \widehat{\mathcal{L}}}{\partial \boldsymbol{z_j}} - \boldsymbol{1}\mu_g \right), \frac{\hat{\boldsymbol{y}}_j}{||\hat{\boldsymbol{y}}_j||} \right\rangle \right\|^2 \tag{7}$$

$$= \frac{\gamma^2}{\sigma^2} \left( \left\| \left( \frac{\partial \widehat{\mathcal{L}}}{\partial \boldsymbol{z_j}} - \boldsymbol{1}\mu_g \right) \right\|^2 - \left\langle \left( \frac{\partial \widehat{\mathcal{L}}}{\partial \boldsymbol{z_j}} - \boldsymbol{1}\mu_g \right), \frac{\hat{\boldsymbol{y}}_j}{||\hat{\boldsymbol{y}}_j||} \right\rangle^2 \right) \tag{8}$$

$$= \frac{\gamma^2}{\sigma^2} \left( \left\| \frac{\partial \widehat{\mathcal{L}}}{\partial \boldsymbol{z_j}} \right\|^2 - \frac{1}{m} \left\langle \boldsymbol{1}, \frac{\partial \widehat{\mathcal{L}}}{\partial \boldsymbol{z_j}} \right\rangle^2 - \frac{1}{\sqrt{m}} \left\langle \frac{\partial \widehat{\mathcal{L}}}{\partial \boldsymbol{z_j}}, \hat{\boldsymbol{y}}_j \right\rangle^2 \right) \tag{9}$$

Exploiting the fact that $\partial \widehat{\mathcal{L}}/\partial z_j = \partial \mathcal{L}/\partial y$ gives the desired result. $\qquad\square$

Next, we can use this to prove the minimax bound on the Lipschitzness with respect to the weights.

**Theorem 4.4** (Minimax bound on weight-space Lipschitzness). *For a BatchNorm network with loss $\widehat{\mathcal{L}}$ and an identical non-BN network (with identical loss $\mathcal{L}$), if*

$$g_j = \max_{||X|| \leq \lambda} ||\nabla_W \mathcal{L}||^2, \qquad \hat{g}_j = \max_{||X|| \leq \lambda} \left\| \nabla_W \widehat{\mathcal{L}} \right\|^2 \implies \hat{g}_j \leq \frac{\gamma^2}{\sigma_j^2} \left( g_j^2 - m\mu_{g_j}^2 - \lambda^2 \left\langle \nabla_{\boldsymbol{y_j}} \mathcal{L}, \hat{\boldsymbol{y}}_j \right\rangle^2 \right).$$

*Proof.* To prove this, we start with the following identity for the largest eigenvalue $\lambda_0$ of $M \in \mathbb{R}^{d \times d}$:

$$\lambda_0 = \max_{x \in \mathbb{R}^d; ||x||_2 = 1} x^\top M x, \tag{10}$$

which in turn implies that for a matrix $X$ with $||X||_2 \leq \lambda$, it must be that $v^\top X v \leq \lambda ||v||^2$, with the choice of $X = \lambda I$ making this bound tight.

Now, we derive the gradient with respect to the weights via the chain rule:

$$\frac{\partial\widehat{\mathcal{L}}}{\partial W_{ij}} = \sum_{b=1}^{m} \frac{\partial\widehat{\mathcal{L}}}{\partial y_j^{(b)}} \frac{\partial y_j^{(b)}}{\partial W_{ij}} \tag{11}$$

$$\frac{\partial\widehat{\mathcal{L}}}{\partial W_{ij}} = \sum_{b=1}^{m} \frac{\partial\widehat{\mathcal{L}}}{\partial y_j^{(b)}} x_i^{(b)} \tag{12}$$

$$= \left\langle \frac{\partial\widehat{\mathcal{L}}}{\partial \boldsymbol{y_j}}, \boldsymbol{x_i} \right\rangle \tag{13}$$

$$\frac{\partial\widehat{\mathcal{L}}}{\partial W_{\cdot j}} = \boldsymbol{X}^\top \left( \frac{\partial\widehat{\mathcal{L}}}{\partial \boldsymbol{y_j}} \right), \tag{14}$$

where $\boldsymbol{X} \in \mathbb{R}^{m \times d}$ is the input matrix holding $X_{bi} = x_i^{(b)}$. Thus,

$$\left\| \frac{\partial\widehat{\mathcal{L}}}{\partial W_{\cdot j}} \right\|^2 = \left( \frac{\partial\widehat{\mathcal{L}}}{\partial \boldsymbol{y_j}} \right)^\top \boldsymbol{X}\boldsymbol{X}^\top \left( \frac{\partial\widehat{\mathcal{L}}}{\partial \boldsymbol{y_j}} \right), \tag{15}$$

and since we have $||X||_2 \le \lambda$, we must have $||XX^\top||_2 \le \lambda^2$, and so recalling (10),

$$\max_{||X||_2 < \lambda} \left\| \frac{\partial\widehat{\mathcal{L}}}{\partial W_{\cdot j}} \right\|^2 \le \lambda^2 \left( \frac{\partial\widehat{\mathcal{L}}}{\partial \boldsymbol{y_j}} \right)^\top \left( \frac{\partial\widehat{\mathcal{L}}}{\partial \boldsymbol{y_j}} \right) = \lambda^2 \left\| \frac{\partial\widehat{\mathcal{L}}}{\partial \boldsymbol{y_j}} \right\|^2, \tag{16}$$

and applying Theorem 4.1 yields:

$$\hat{g}_j := \max_{||X||_2 < \lambda} \left\| \frac{\partial\widehat{\mathcal{L}}}{\partial W_{\cdot j}} \right\|^2 \le \frac{\lambda^2 \gamma^2}{\sigma^2} \left( \left\| \frac{\partial\mathcal{L}}{\partial \boldsymbol{y_j}} \right\|^2 - \frac{1}{m} \left\langle \mathbf{1}, \frac{\partial\mathcal{L}}{\partial \boldsymbol{y_j}} \right\rangle^2 - \frac{1}{\sqrt{m}} \left\langle \frac{\partial\mathcal{L}}{\partial \boldsymbol{y_j}}, \hat{\boldsymbol{y}}_j \right\rangle^2 \right). \tag{17}$$

Finally, by applying (10) again, note that in fact in the normal network,

$$g_j := \max_{||X||_2 < \lambda} \left\| \frac{\partial\widehat{\mathcal{L}}}{\partial W_{\cdot j}} \right\|^2 = \lambda^2 \left\| \frac{\partial\mathcal{L}}{\partial \boldsymbol{y_j}} \right\|^2, \tag{18}$$

and thus

$$\hat{g}_j \le \frac{\gamma^2}{\sigma^2} \left( g_j^2 - m\mu_{g_j}^2 - \lambda^2 \left\langle \frac{\partial\mathcal{L}}{\partial \boldsymbol{y_j}}, \hat{\boldsymbol{y}}_j \right\rangle^2 \right).$$

$\square$

**Theorem 4.2** (The effect of BN to smoothness). *Let $\hat{\boldsymbol{g}}_j = \nabla_{\boldsymbol{y}_j}\mathcal{L}$ and $\boldsymbol{H}_{jj} = \frac{\partial \mathcal{L}}{\partial \boldsymbol{y}_j \partial \boldsymbol{y}_j}$ be the gradient and Hessian of the loss with respect to the layer outputs respectively. Then*

$$\left(\nabla_{\boldsymbol{y}_j}\widehat{\mathcal{L}}\right)^\top \frac{\partial \widehat{\mathcal{L}}}{\partial \boldsymbol{y}_j \partial \boldsymbol{y}_j} \left(\nabla_{\boldsymbol{y}_j}\widehat{\mathcal{L}}\right) \leq \frac{\gamma^2}{\sigma^2}\left(\frac{\partial \widehat{\mathcal{L}}}{\partial \boldsymbol{y}_j}\right)^\top \boldsymbol{H}_{jj}\left(\frac{\partial \widehat{\mathcal{L}}}{\partial \boldsymbol{y}_j}\right) - \frac{\gamma}{m\sigma^2}\langle \hat{\boldsymbol{g}}_j, \hat{\boldsymbol{y}}_j \rangle \left\|\frac{\partial \widehat{\mathcal{L}}}{\partial \boldsymbol{y}_j}\right\|^2$$

*If we also have that the $\boldsymbol{H}_{jj}$ preserves the relative norms of $\hat{\boldsymbol{g}}_j$ and $\nabla_{\boldsymbol{y}_j}\widehat{\mathcal{L}}$,*

$$\left(\nabla_{\boldsymbol{y}_j}\widehat{\mathcal{L}}\right)^\top \frac{\partial \widehat{\mathcal{L}}}{\partial \boldsymbol{y}_j \partial \boldsymbol{y}_j} \left(\nabla_{\boldsymbol{y}_j}\widehat{\mathcal{L}}\right) \leq \frac{\gamma^2}{\sigma^2}\left(\hat{\boldsymbol{g}}_j^\top \boldsymbol{H}_{jj}\hat{\boldsymbol{g}}_j - \frac{1}{m\gamma}\langle \hat{\boldsymbol{g}}_j, \hat{\boldsymbol{y}}_j \rangle \left\|\frac{\partial \widehat{\mathcal{L}}}{\partial \boldsymbol{y}_j}\right\|^2\right)$$

*Proof.* We use the following notation freely in the following. First, we introduce the hessian with respect to the final activations as:

$$\boldsymbol{H}_{jk} \in \mathbb{R}^{m \times m}; H_{jk} := \frac{\partial \widehat{\mathcal{L}}}{\partial \boldsymbol{z}_j \partial \boldsymbol{z}_k} = \frac{\partial \mathcal{L}}{\partial \boldsymbol{y}_j \partial \boldsymbol{y}_k},$$

where the final equality is by the assumptions of our setup. Once again for convenience, we define a function $\mu_{(\cdot)}$ which operates on vectors and matrices and gives their element-wise mean; in particular, $\mu_{(\boldsymbol{v})} = \frac{1}{d}\mathbf{1}^\top v$ for $v \in \mathbb{R}^d$ and we write $\boldsymbol{\mu}_{(\cdot)} = \mu_{(\cdot)}\mathbf{1}$ to be a vector with all elements equal to $\mu$. Finally, we denote the gradient with respect to the batch-normalized outputs as $\hat{\boldsymbol{g}}_j$, such that:

$$\hat{\boldsymbol{g}}_j = \frac{\partial \widehat{\mathcal{L}}}{\partial \boldsymbol{z}_j} = \frac{\partial \mathcal{L}}{\partial \boldsymbol{y}_j},$$

where again the last equality is by assumption.

Now, we begin by looking at the Hessian of the loss with respect to the pre-BN activations $\boldsymbol{y}_j$ using the expanded gradient as above:

$$\frac{\partial \widehat{\mathcal{L}}}{\partial \boldsymbol{y}_j \partial \boldsymbol{y}_j} = \frac{\partial}{\partial \boldsymbol{y}_j}\left(\left(\frac{\gamma}{m\sigma_j}\right)\left[m\hat{\boldsymbol{g}}_j - m\boldsymbol{\mu}_{(\hat{\boldsymbol{g}}_j)} - \hat{y}_j^{(b)}\langle \hat{\boldsymbol{g}}_j, \hat{\boldsymbol{y}}_j \rangle\right]\right) \tag{19}$$

Using the product rule and the chain rule:

$$= \frac{\gamma}{m\sigma}\left(\frac{\partial}{\partial \boldsymbol{z}_q}\left[m\hat{\boldsymbol{g}}_j - m\boldsymbol{\mu}_{(\hat{\boldsymbol{g}}_j)} - \hat{\boldsymbol{y}}_j\langle \hat{\boldsymbol{g}}_j, \hat{\boldsymbol{y}}_j \rangle\right]\right) \cdot \frac{\partial \boldsymbol{z}_q}{\partial \boldsymbol{y}_j} \tag{20}$$

$$+ \left(\frac{\partial}{\partial \boldsymbol{y}_j}\left(\frac{\gamma}{m\sigma_j}\right)\right) \cdot \left(m\hat{\boldsymbol{g}}_j - m\boldsymbol{\mu}_{(\hat{\boldsymbol{g}}_j)} - \hat{\boldsymbol{y}}_j\langle \hat{\boldsymbol{g}}_j, \hat{\boldsymbol{y}}_j \rangle\right) \tag{21}$$

Distributing the derivative across subtraction:

$$= \left(\frac{\gamma}{\sigma_j}\right)\left(\boldsymbol{H}_{jj} - \frac{\partial \boldsymbol{\mu}_{(\hat{\boldsymbol{g}}_j)}}{\partial \boldsymbol{z}_j} - \frac{\partial}{\partial \boldsymbol{z}_j}\left(\frac{1}{m}\hat{\boldsymbol{y}}_j\langle \hat{\boldsymbol{g}}_j, \hat{\boldsymbol{y}}_j \rangle\right)\right) \cdot \frac{\partial \boldsymbol{z}_j}{\partial \boldsymbol{y}_j} \tag{22}$$

$$+ \left(\hat{\boldsymbol{g}}_j - \boldsymbol{\mu}_{(\hat{\boldsymbol{g}}_j)} - \frac{1}{m}\hat{\boldsymbol{y}}_j\langle \hat{\boldsymbol{g}}_j, \hat{\boldsymbol{y}}_j \rangle\right)\left(\frac{\partial}{\partial \boldsymbol{y}_j}\left(\frac{\gamma}{\sigma_j}\right)\right) \tag{23}$$

We address each of the terms in the above (22) and (23) one by one:

$$\frac{\partial \boldsymbol{\mu}_{(\hat{\boldsymbol{g}}_j)}}{\partial \boldsymbol{z}_j} = \frac{1}{m}\frac{\partial \mathbf{1}^\top \hat{\boldsymbol{g}}_j}{\partial \boldsymbol{z}_j} = \frac{1}{m}\mathbf{1} \cdot \mathbf{1}^\top \boldsymbol{H}_{jj} \tag{24}$$

$$\frac{\partial}{\partial \boldsymbol{z_j}} \left( \boldsymbol{\hat{y}}_j \left\langle \boldsymbol{\hat{y}}_j, \boldsymbol{\hat{g}_j} \right\rangle \right) = \frac{1}{\gamma} \frac{\partial}{\partial \boldsymbol{\hat{y}}_j} \left( \boldsymbol{\hat{y}}_j \left\langle \boldsymbol{\hat{y}}_j, \boldsymbol{\hat{g}_j} \right\rangle \right) \tag{25}$$

$$= \frac{1}{\gamma} \frac{\partial \boldsymbol{\hat{y}}_j}{\partial \boldsymbol{\hat{y}}_j} \left\langle \boldsymbol{\hat{g}_j}, \boldsymbol{\hat{y}_j} \right\rangle + \boldsymbol{\hat{y}}_j \boldsymbol{\hat{y}}_j^\top \boldsymbol{H}_{jj} + \frac{1}{\gamma} \boldsymbol{\hat{y}}_j \boldsymbol{\hat{g}}_j^\top \frac{\partial \boldsymbol{\hat{y}}_j}{\partial \boldsymbol{\hat{y}}_j} \tag{26}$$

$$= \frac{1}{\gamma} \boldsymbol{I} \left\langle \boldsymbol{\hat{g}_j}, \boldsymbol{\hat{y}_j} \right\rangle + \boldsymbol{\hat{y}}_j \boldsymbol{\hat{y}}_j^\top \boldsymbol{H}_{jj} + \frac{1}{\gamma} \boldsymbol{\hat{y}}_j \boldsymbol{\hat{g}}_j^\top \boldsymbol{I} \tag{27}$$

$$\frac{\partial}{\partial \boldsymbol{y_j}} \left( \frac{\gamma}{\sigma_j} \right) = \gamma \sqrt{m} \frac{\partial \left( \left( \boldsymbol{y_j} - \boldsymbol{\mu}_{(\boldsymbol{y_j})} \right)^\top \left( \boldsymbol{y_j} - \boldsymbol{\mu}_{(\boldsymbol{y_j})} \right) \right)^{-\frac{1}{2}}}{\partial \boldsymbol{y_j}} \tag{28}$$

$$= \frac{-1}{2} \gamma \sqrt{m} \left( \left( \boldsymbol{y_j} - \boldsymbol{\mu}_{(\boldsymbol{y_j})} \right)^\top \left( \boldsymbol{y_j} - \boldsymbol{\mu}_{(\boldsymbol{y_j})} \right) \right)^{-\frac{3}{2}} \left( 2 \left( \boldsymbol{y_j} - \boldsymbol{\mu}_{(\boldsymbol{y_j})} \right) \right) \tag{29}$$

$$= -\frac{\gamma}{m\sigma^2} \boldsymbol{\hat{y}}_j \tag{30}$$

Now, we can use the preceding to rewrite the Hessian as:

$$\frac{\partial \widehat{\mathcal{L}}}{\partial \boldsymbol{y_j} \partial \boldsymbol{y_j}} = \left( \frac{\gamma}{m\sigma_j} \right) \left( m \boldsymbol{H}_{jj} - \boldsymbol{1} \cdot \boldsymbol{1}^\top \boldsymbol{H}_{jj} - \frac{1}{\gamma} \boldsymbol{I} \left\langle \boldsymbol{\hat{g}_j}, \boldsymbol{\hat{y}_j} \right\rangle - \boldsymbol{\hat{y}}_j \boldsymbol{\hat{y}}_j^\top \boldsymbol{H}_{jj} - \frac{1}{\gamma} \left( \boldsymbol{\hat{y}}_j \boldsymbol{\hat{g}}_j^\top \right) \right) \cdot \frac{\partial \boldsymbol{z_j}}{\partial \boldsymbol{y_j}} \tag{31}$$

$$- \frac{\gamma}{m\sigma^2} \left( \boldsymbol{\hat{g}_j} - \boldsymbol{\mu}_{(\boldsymbol{\hat{g}_j})} - \frac{1}{m} \boldsymbol{\hat{y}}_j \left\langle \boldsymbol{\hat{g}_j}, \boldsymbol{\hat{y}_j} \right\rangle \right) \boldsymbol{\hat{y}}_j^\top \tag{32}$$

Now, using Fact C.2, we have that:

$$\frac{\partial \boldsymbol{z_j}}{\partial \boldsymbol{y_j}} = \left( \frac{\gamma}{\sigma_j} \right) \left( \boldsymbol{I} - \frac{1}{m} \boldsymbol{1} \cdot \boldsymbol{1}^\top - \frac{1}{m} \boldsymbol{\hat{y}}_j \boldsymbol{\hat{y}}_j^\top \right), \tag{33}$$

and substituting this yields (letting $\boldsymbol{M} = \boldsymbol{1} \cdot \boldsymbol{1}^\top$ for convenience):

$$\frac{\partial \widehat{\mathcal{L}}}{\partial \boldsymbol{y_j} \partial \boldsymbol{y_j}} = \frac{\gamma^2}{m\sigma^2} \left( m \boldsymbol{H}_{jj} - \boldsymbol{M} \boldsymbol{H}_{jj} - \frac{1}{\gamma} \boldsymbol{I} \left\langle \boldsymbol{\hat{g}_j}, \boldsymbol{\hat{y}_j} \right\rangle - \boldsymbol{\hat{y}}_j \boldsymbol{\hat{y}}_j^\top \boldsymbol{H}_{jj} - \frac{1}{\gamma} \left( \boldsymbol{\hat{y}}_j \boldsymbol{\hat{g}}_j^\top \right) \right) \tag{34}$$

$$- \frac{\gamma^2}{m\sigma^2} \left( \boldsymbol{H}_{jj} \boldsymbol{M} - \frac{1}{m} \boldsymbol{M} \boldsymbol{H}_{jj} \boldsymbol{M} - \frac{1}{m\gamma} \boldsymbol{M} \left\langle \boldsymbol{\hat{g}_j}, \boldsymbol{\hat{y}_j} \right\rangle - \frac{1}{m} \boldsymbol{\hat{y}}_j \boldsymbol{\hat{y}}_j^\top \boldsymbol{H}_{jj} \boldsymbol{M} - \frac{1}{m\gamma} \left( \boldsymbol{\hat{y}}_j \boldsymbol{\hat{g}}_j^\top \boldsymbol{M} \right) \right) \tag{35}$$

$$- \frac{\gamma^2}{m\sigma^2} \left( \boldsymbol{H}_{jj} \boldsymbol{\hat{y}}_j \boldsymbol{\hat{y}}_j^\top - \frac{1}{m} \boldsymbol{M} \boldsymbol{H}_{jj} \boldsymbol{\hat{y}}_j \boldsymbol{\hat{y}}_j^\top - \frac{1}{m\gamma} \boldsymbol{\hat{y}}_j \boldsymbol{\hat{y}}_j^\top \left\langle \boldsymbol{\hat{g}_j}, \boldsymbol{\hat{y}_j} \right\rangle - \frac{1}{m} \boldsymbol{\hat{y}}_j \boldsymbol{\hat{y}}_j^\top \boldsymbol{H}_{jj} \boldsymbol{\hat{y}}_j \boldsymbol{\hat{y}}_j^\top - \frac{1}{m\gamma} \left( \boldsymbol{\hat{y}}_j \boldsymbol{\hat{g}}_j^\top \boldsymbol{\hat{y}}_j \boldsymbol{\hat{y}}_j^\top \right) \right) \tag{36}$$

$$- \frac{\gamma}{m\sigma^2} \left( \boldsymbol{\hat{g}_j} - \boldsymbol{\mu}_{(\boldsymbol{\hat{g}_j})} - \frac{1}{m} \boldsymbol{\hat{y}}_j \left\langle \boldsymbol{\hat{g}_j}, \boldsymbol{\hat{y}_j} \right\rangle \right) \boldsymbol{\hat{y}}_j^\top \tag{37}$$

Collecting the terms, and letting $\overline{\boldsymbol{\hat{g}_j}} = \boldsymbol{\hat{g}_j} - \boldsymbol{\mu}_{(\boldsymbol{\hat{g}_j})}$:

$$\frac{\partial \widehat{\mathcal{L}}}{\partial \boldsymbol{y_j} \partial \boldsymbol{y_j}} = \frac{\gamma^2}{m\sigma^2} \Big[ m \boldsymbol{H}_{jj} - \boldsymbol{M} \boldsymbol{H}_{jj} - \boldsymbol{\hat{y}}_j \boldsymbol{\hat{y}}_j^\top \boldsymbol{H}_{jj} - \boldsymbol{H}_{jj} \boldsymbol{M} + \frac{1}{m} \boldsymbol{M} \boldsymbol{H}_{jj} \boldsymbol{M} \tag{38}$$

$$+ \frac{1}{m} \boldsymbol{\hat{y}}_j \boldsymbol{\hat{y}}_j^\top \boldsymbol{H}_{jj} \boldsymbol{M} - \boldsymbol{H}_{jj} \boldsymbol{\hat{y}}_j \boldsymbol{\hat{y}}_j^\top + \frac{1}{m} \boldsymbol{M} \boldsymbol{H}_{jj} \boldsymbol{\hat{y}}_j \boldsymbol{\hat{y}}_j^\top + \frac{1}{m} \boldsymbol{\hat{y}}_j \boldsymbol{\hat{y}}_j^\top \boldsymbol{H}_{jj} \boldsymbol{\hat{y}}_j \boldsymbol{\hat{y}}_j^\top \Big] \tag{39}$$

$$- \frac{\gamma}{m\sigma^2} \left( \boldsymbol{\hat{g}_j} \boldsymbol{\hat{y}}_j^\top - \boldsymbol{\mu}_{(\boldsymbol{\hat{g}_j})} \boldsymbol{\hat{y}}_j^\top - \frac{3}{m} \boldsymbol{\hat{y}}_j \boldsymbol{\hat{y}}_j^\top \left\langle \boldsymbol{\hat{g}_j}, \boldsymbol{\hat{y}_j} \right\rangle + \left( \left\langle \boldsymbol{\hat{g}_j}, \boldsymbol{\hat{y}_j} \right\rangle \boldsymbol{I} + \boldsymbol{\hat{y}}_j \boldsymbol{\hat{g}}_j^\top \right) \left( \boldsymbol{I} - \frac{1}{m} \boldsymbol{M} \right) \right) \tag{40}$$

$$= \frac{\gamma^2}{\sigma^2} \Bigg[ \left( \boldsymbol{I} - \frac{1}{m} \boldsymbol{\hat{y}}_j \boldsymbol{\hat{y}}_j^\top - \frac{1}{m} \boldsymbol{M} \right) \boldsymbol{H}_{jj} \left( \boldsymbol{I} - \frac{1}{m} \boldsymbol{\hat{y}}_j \boldsymbol{\hat{y}}_j^\top - \frac{1}{m} \boldsymbol{M} \right) \tag{41}$$

$$- \frac{1}{m\gamma} \left( \overline{\boldsymbol{\hat{g}_j}} \boldsymbol{\hat{y}}_j^\top + \boldsymbol{\hat{y}}_j \overline{\boldsymbol{\hat{g}}}_j^\top - \frac{3}{m} \boldsymbol{\hat{y}}_j \boldsymbol{\hat{g}}_j^\top \boldsymbol{\hat{y}}_j \boldsymbol{\hat{y}}_j^\top + \left\langle \boldsymbol{\hat{g}_j}, \boldsymbol{\hat{y}_j} \right\rangle \left( \boldsymbol{I} - \frac{1}{m} \boldsymbol{M} \right) \right) \Bigg] \tag{42}$$

Now, we wish to calculate the effective beta smoothness with respect to a batch of activations, which corresponds to $g^\top H g$, where $g$ is the gradient with respect to the activations (as derived in the previous proof). We expand this product noting the following identities:

$$M\overline{\hat{g}_j} = 0 \tag{43}$$

$$\left(I - \frac{1}{m}M - \frac{1}{m}\hat{y}_j\hat{y}_j^\top\right)^2 = \left(I - \frac{1}{m}M - \frac{1}{m}\hat{y}_j\hat{y}_j^\top\right) \tag{44}$$

$$\hat{y}_j^\top\left(I - \frac{1}{m}\hat{y}_j\hat{y}_j^\top\right) = \mathbf{0} \tag{45}$$

$$\left(I - \frac{1}{m}M\right)\left(I - \frac{1}{m}\hat{y}_j\hat{y}_j^\top\right)\overline{\hat{g}_j} = \left(I - \frac{1}{m}\hat{y}_j\hat{y}_j^\top\right)\overline{\hat{g}_j} \tag{46}$$

Also recall from (5) that:

$$\frac{\partial\widehat{\mathcal{L}}}{\partial y_j} = \frac{\gamma}{\sigma}\overline{\hat{g}_j}^\top\left(I - \frac{1}{m}\hat{y}_j\hat{y}_j^\top\right) \tag{47}$$

Applying these while expanding the product gives:

$$\frac{\partial\widehat{\mathcal{L}}}{\partial y_j}^\top \cdot \frac{\partial\widehat{\mathcal{L}}}{\partial y_j\partial y_j} \cdot \frac{\partial\widehat{\mathcal{L}}}{\partial y_j} = \frac{\gamma^4}{\sigma^4}\overline{\hat{g}_j}^\top\left(I - \frac{1}{m}\hat{y}_j\hat{y}_j^\top\right)H_{jj}\left(I - \frac{1}{m}\hat{y}_j\hat{y}_j^\top\right)\overline{\hat{g}_j} \tag{48}$$

$$- \frac{\gamma^3}{m\sigma^4}\overline{\hat{g}_j}^\top\left(I - \frac{1}{m}\hat{y}_j\hat{y}_j^\top\right)\overline{\hat{g}_j}\langle\hat{g}_j, \hat{y}_j\rangle \tag{49}$$

$$= \frac{\gamma^2}{\sigma^2}\left(\frac{\partial\widehat{\mathcal{L}}}{\partial y_j}\right)^\top H_{jj}\left(\frac{\partial\widehat{\mathcal{L}}}{\partial y_j}\right) - \frac{\gamma}{m\sigma^2}\langle\hat{g}_j, \hat{y}_j\rangle\left\|\frac{\partial\widehat{\mathcal{L}}}{\partial y_j}\right\|^2 \tag{50}$$

This concludes the first part of the proof. Note that if $H_{jj}$ preserves the relative norms of $\hat{g}_j$ and $\nabla_{y_j}\widehat{\mathcal{L}}$, then the final statement follows trivially, since the first term of the above is simply the induced squared norm $\left\|\frac{\partial\widehat{\mathcal{L}}}{\partial y_j}\right\|_{H_{jj}}^2$, and so

$$\frac{\partial\widehat{\mathcal{L}}}{\partial y_j}^\top \cdot \frac{\partial\widehat{\mathcal{L}}}{\partial y_j\partial y_j} \cdot \frac{\partial\widehat{\mathcal{L}}}{\partial y_j} \leq \frac{\gamma^2}{\sigma^2}\left[\hat{g}_j^\top H_{jj}\hat{g}_j - \frac{1}{m\gamma}\langle\hat{g}_j, \hat{y}_j\rangle\left\|\frac{\partial\widehat{\mathcal{L}}}{\partial y_j}\right\|^2\right] \tag{51}$$

$\square$

Once again, the same techniques also give us a minimax separation:

**Theorem C.1** (Minimax smoothness bound). *Under the same conditions as the previous theorem,*

$$\max_{||X||\leq\lambda}\left(\frac{\partial\widehat{\mathcal{L}}}{\partial W_{\cdot j}}\right)^\top \frac{\partial\widehat{\mathcal{L}}}{\partial W_{\cdot j}\partial W_{\cdot j}}\left(\frac{\partial\widehat{\mathcal{L}}}{\partial W_{\cdot j}}\right) < \frac{\gamma^2}{\sigma^2}\left[\max_{||X||\leq\lambda}\left(\frac{\partial\mathcal{L}}{\partial W_{\cdot j}}\right)^\top \frac{\partial\mathcal{L}}{\partial W_{\cdot j}\partial W_{\cdot j}}\left(\frac{\partial\mathcal{L}}{\partial W_{\cdot j}}\right) - \lambda^4\kappa\right],$$

*where $\kappa$ is the separation given in the previous theorem.*

*Proof.*

$$\frac{\partial\mathcal{L}}{\partial W_{ij}\partial W_{kj}} = x_i^\top\frac{\partial\mathcal{L}}{\partial y_j\partial y_j}x_k \tag{52}$$

$$\frac{\partial\widehat{\mathcal{L}}}{\partial W_{ij}\partial W_{kj}} = x_i^\top\frac{\partial\widehat{\mathcal{L}}}{\partial y_j\partial y_j}x_k \tag{53}$$

$$\frac{\partial\widehat{\mathcal{L}}}{\partial W_{\cdot j}\partial W_{\cdot j}} = X^\top\frac{\partial\widehat{\mathcal{L}}}{\partial y_j\partial y_j}X \tag{54}$$

$$\tag{55}$$

Looking at the gradient predictiveness using the gradient we derived in the first proofs:

$$\beta := \left(\frac{\partial\widehat{\mathcal{L}}}{\partial W_{\cdot j}}\right)^{\top} \frac{\partial\widehat{\mathcal{L}}}{\partial W_{\cdot j}\partial W_{\cdot j}} \left(\frac{\partial\widehat{\mathcal{L}}}{\partial W_{\cdot j}}\right) \tag{56}$$

$$= \hat{\boldsymbol{g}}_{\boldsymbol{j}}^{\top} \left(\boldsymbol{I} - \frac{1}{m}\hat{\boldsymbol{y}}_j\hat{\boldsymbol{y}}_j^{\top}\right) \boldsymbol{X}\boldsymbol{X}^{\top} \frac{\partial\widehat{\mathcal{L}}}{\partial\boldsymbol{y_j}\partial\boldsymbol{y_j}} \boldsymbol{X}\boldsymbol{X}^{\top} \left(\boldsymbol{I} - \frac{1}{m}\hat{\boldsymbol{y}}_j\hat{\boldsymbol{y}}_j^{\top}\right) \hat{\boldsymbol{g}}_{\boldsymbol{j}} \tag{57}$$

Maximizing the norm with respect to $X$ yields:

$$\max_{||X||\leq\lambda} \beta = \lambda^4 \hat{\boldsymbol{g}}_{\boldsymbol{j}}^{\top} \left(\boldsymbol{I} - \frac{1}{m}\hat{\boldsymbol{y}}_j\hat{\boldsymbol{y}}_j^{\top}\right) \frac{\partial\widehat{\mathcal{L}}}{\partial\boldsymbol{y_j}\partial\boldsymbol{y_j}} \left(\boldsymbol{I} - \frac{1}{m}\hat{\boldsymbol{y}}_j\hat{\boldsymbol{y}}_j^{\top}\right) \hat{\boldsymbol{g}}_{\boldsymbol{j}}, \tag{58}$$

at which the previous proof can be applied to conclude. $\qquad\square$

**Lemma 4.5** (BatchNorm leads to a favourable initialization). *Let $\boldsymbol{W^*}$ and $\widehat{\boldsymbol{W}}^{\boldsymbol{*}}$ be the set of local optima for the weights in the normal and BN networks, respectively. For any initialization $W_0$*

$$\left|\left|W_0 - \widehat{W}^*\right|\right|^2 \leq ||W_0 - W^*||^2 - \frac{1}{||W^*||^2}\left(||W^*||^2 - \langle W^*, W_0\rangle\right)^2,$$

*if $\langle W_0, W^*\rangle > 0$, where $\widehat{W}^*$ and $W^*$ are closest optima for BN and standard network, respectively.*

*Proof.* This is as a result of the scale-invariance of batch normalization. In particular, first note that for any optimum $W$ in the standard network, we have that any scalar multiple of $W$ must also be an optimum in the BN network (since $BN((aW)x) = BN(Wx)$ for all $a > 0$). Recall that we have defined $k > 0$ to be propertial to the correlation between $W_0$ and $W^*$:

$$k = \frac{\langle W^*, W_0\rangle}{||W^*||^2}$$

Thus, for any optimum $W^*$, we must have that $\widehat{W} := kW^*$ must be an optimum in the BN network. The difference between distance to this optimum and the distance to $W$ is given by:

$$\left|\left|W_0 - \widehat{W}\right|\right|^2 - ||W_0 - W^*||^2 = ||W_0 - kW^*||^2 - ||W_0 - W^*||^2 \tag{59}$$

$$= \left(||W_0||^2 - k^2\,||W^*||^2\right) - \left(||W_0||^2 - 2k\,||W^*||^2 + ||W^*||^2\right) \tag{60}$$

$$= 2k\,||W^*||^2 - k^2\,||W^*||^2 - ||W^*||^2 \tag{61}$$

$$= -\,||W^*||^2 \cdot (1-k)^2 \tag{62}$$

$\square$

## Footnotes

[4]We choose to not experiment with ResNets [7] since they seem to provide several similar benefits to BatchNorm [6] and would introduce conflating factors into our study.

[5]While the factorized formulation is equivalent to a single matrix in terms of expressivity, the optimization landscape is drastically different [6].