[Reviews · NeurIPS 2018]

Reviewer 1



This is a very interesting piece of work, challenging a preconception that a lot of deep learning practitioners have. The authors show that batch norm has very little effect on "internal covariate shift". They postulate that its benefit is due to a smoothing effect on the optimisation landscape. As far as I am aware, there is very little work exploring the underlying mechanisms of batch norm. The paper is easy to follow, and well written with a clear narrative. Having a theoretical analysis alongside an empirical one is encouraging. A few comments: - 3D plots (in figures 1 and 2) are fairly difficult to interpret (beyond observing that the width changes) - What explicitly is the loss gradient in figure 4? Is it simply the vector of dLoss/dWeights for the network? - Have you observed the effect of batch norm on more modern networks (e.g. Resnets/Densenets?) ------------------------------ Post-rebuttal ------------------------------ Having read the rebuttal, and the other reviews I am happy with my verdict (7)

Reviewer 2



Batch normalization accelerates the training process of deep networks and has become de facto standard. Despite the widespread popularity, we have little evidence of where such improvement comes from. This paper explores the roots of success for batch normalization, which was believed to the reduction of internal covariate shift phenomenon, as suggested by the original batch normalization paper. This paper argues it is not because batch norm reduces internal covariate shift. Instead, it is because batch norm smoothens the loss gradient landscape such that gradient is more predictable even with large step sizes. My main question is regarding Section 2.2 “Is BatchNorm reducing internal covariate shift?”. I was a bit confused about the new definition of internal covariate shift. In the original paper, by internal covariate shift, they refer to the changes of each layer’s input distribution. In this sense, the reduction of internal covariate shift appears to be one of the simple consequences of batch norm’s mechanics. But when we adopt the new definition, the internal covariate shift goes up as we optimize. I can see the new definition being more convenient from an optimization perspective, but perhaps it is a bit strange to conclude batch norm does not reduce internal covariate shift at all from empirical results based on a different definition. Despite so, the results based on the new definition are still interesting to think about. As suggested in the end, the smoothening effect of batch norm could encourage the training process to converge to more flat minima. I am curious if other normalization strategies, which aim at the same smoothening effect, also tend to improve generalization as well. A minor concern: Activation histograms in Figure 2 and Figure 7: It is a bit hard for me to interpret such histograms without the explanation of what each axis means.

Reviewer 3



Summary The authors present a new justification of the effectiveness of Batch Normalization. They show empirically that what was previously believed to be the main advantage of Batch Normalization, i.e. reducing Internal Covariate Shifts, is actually not something that is modified by this reparametrization and may even be amplified by it. In light of this, they present a new analysis using lipschitz smoothness and how it stabilises both loss landspace and gradients leading to a better (or easier) optimization. This work is very much needed for a better understanding of the underlying effects of batch normalization on optimization and generatlization and it is a natural extension of the recent literature of theoretical deep learning. Quality: The empirical proof on the internal covariate shift is too qualitative. The Figure 1 gives at best a rough idea of what is going on but there is no quantititave measure of the rate of change of the moments of the distribution for instance. Subsection 2.2 is a good attempt however at quantifying the shift. The analysis of section 3 falls somehow short. A strong relation between the lipschitz smoothness and optimization should be explicitly given in the paper. The empricial analysis is not broad enough in my opinion. As listed in this paper, there is a plethora of alternatives to batch-normalization that have been proposed in the past years. Since they all have their up and downs, comparing them on the metrics of ICS as proposed in this paper or lipschitz smoothness would most probably be enligthning. Clarity: The paper is well written and concise. The storyline is clear and easy to follow. Originality: As far as I know this is the first analysis to shed light on the stability and smoothness effect of batch normalization as its most important feature for helping optimization. Significance: This paper is important for two reasons. 1) Batch Normalization is a widely used technique, yet its important underlying properties are not very well understood. 2) Since Batch Normalization seems to help with both optimization and generatlization, a better understanding of it could give fundamental insights for the subfield of theoretical deep learning.